# Preliminary Evaluation of Self-Reported Training Volume as an Adjunct Measure of Female Athlete Triad Risk in Division 1 Collegiate Female Runners

**DOI:** 10.3390/jfmk9040179

**Published:** 2024-09-27

**Authors:** Sarah Parnell, Austin J. Graybeal, Megan E. Renna, Jon Stavres

**Affiliations:** 1School of Kinesiology and Nutrition, The University of Southern Mississippi, Hattiesburg, MS 39406, USA; sarah.parnell@usm.edu (S.P.); austin.graybeal@usm.edu (A.J.G.); 2School of Psychology, The University of Southern Mississippi, Hattiesburg, MS 39406, USA; megan.renna@usm.edu

**Keywords:** athlete, women, energy availability, exercise volume, RED-S

## Abstract

**Background/Objectives**: This study tested whether self-reported training volume is predictive of female athlete triad risk collected using an established twelve-question triad screening tool in National Collegiate Athletic Association (NCAA) Division I (DI) collegiate female runners. **Methods**: A total of 319 institutions were initially contacted, seven of which agreed to distribute surveys to their female cross-country and track and field athletes. A total of 41 of 149 respondents completed the survey and met all inclusion criteria. Linear and binomial logistic regressions examined the relationships between self-reported training volumes and estimated triad risk. Independent samples *t*-tests were also used to compare training volumes across the high (> 50th percentile for risk factor counts) vs. low-risk groups. **Results**: Total weekly competition and conditioning resistance training hours were associated with the total number of triad risk factors (*p* = 0.044) and were also predictive of the triad risk group (*p* = 0.037). Likewise, both competition and conditioning resistance training hours (*p* = 0.034) were higher in the high-risk group versus the low-risk group. **Conclusions**: These findings suggest that self-reported resistance training volume is predictive of triad risk, but additional research is required to determine if monitoring training volume can provide valuable, real-time assessments of triad risk in DI collegiate female runners.

## 1. Introduction

The female athlete triad is a complex syndrome defined by the co-presentation of distinct subclinical risk factors. These risk factors were initially identified as disordered eating, amenorrhea, and osteoporosis in the early 1990s by the American College of Sports Medicine [1,2]. In 2014, the triad was further categorized under the umbrella of Relative Energy Deficiency in Sports (RED-S [3]), and the definition of triad risk factors has expanded to include the broader components of energy availability (EA), menstrual function, and bone mineral density (BMD), each of which is arranged along a spectrum from optimal health to a diseased state [2,4].

Arguably the most influential component of the triad is energy availability (EA), which has been broadly defined as the amount of energy available for essential physiological functions after accounting for energy expended during physical activity and exercise [2,4]. This net EA is necessary for essential tasks such as growth, homeostasis, and thermoregulation [5,6]. The optimal level of EA for bodily maintenance in adult females has been suggested to be 45 kcal/kg of FFM/day, with greater values required for weight gain and muscle hypertrophy [5]. Low energy availability (LEA), however, is defined as having an EA less than the energy required to sustain the aforementioned physiological functions, or an EA below 30 kcal/kg of FFM/day [5], and may occur with or without disordered eating. In fact, chronic LEA has been identified as the primary factor in the triad due to its overarching contributions to the detrimental reproductive and skeletal health components of the triad [4].

General screening practices for the triad tend to focus on the assessment of an individual’s likelihood of disordered eating habits rather than direct evaluations of the triad components themselves. The screening tool proposed and stratified by the Female Athlete Triad Coalition in 2016 is composed of 12 questions addressing dietary habits, stress injury history, and general menstrual and bone health in order to determine sport participation clearance for female athletes [2]. This assessment method is a popular and convenient choice for triad screening in female athletes because it does not initially require direct measures (e.g., Z-scores for determining low BMD), which ultimately promotes accessibility and broad application. However, this screening tool does not include questions regarding training volume. Given the influence of exercise (also known as, “training”) habits on total daily energy expenditure, it would seem likely that training volume would be a significant contributing factor to the prediction of triad risk. As such, collecting measures of training volume may be a valuable addition to the current triad screening process. However, like BMD scores, collecting objective measures of training volume relies on access to physical activity tracking equipment. This may present a barrier for broad application, whereas, self-reported (e.g., survey-based) measures of training volume could be collected in tandem with the Coalition’s triad screening survey [2], eliminating this potential barrier. Accordingly, this study tested the hypothesis that self-reported measures of training volume would be significantly associated with self-reported indices of triad risk in female collegiate track and field athletes. If true, these findings would support future studies focused on optimizing the collection and monitoring of training volume for female athletes.

## 2. Materials and Methods

### 2.1. Participants and Study Design

This study followed an anonymized, survey-based cross-sectional design, and included only universities with active women’s track and field programs. All participants provided informed consent at the beginning of each survey (via a checkbox), and all study protocols were approved by the University of Southern Mississippi Institutional Review Board (IRB# 22-1474). A total of 319 universities were initially contacted during the distribution process. To qualify for this study, participants had to be biologically female, at least 18 years of age, and participating in an active collegiate women’s National Collegiate Athletic Association (NCAA) Division 1 (D1) track and field program. Participants who were younger than 18 years of age or who indicated that they were pregnant were excluded. Additionally, participants were excluded from the final sample population and data analysis if their response indicated that the survey was closed prior to all questions being reviewed (e.g., answering the first few questions and then submitting the survey without completing any other questions). 3 participants also responded to the survey twice (identified using a unique, self-selected subject identity code). For these participants, only the first response was included in the final analysis. Of the 149 participants who responded to the survey, 41 individuals met all criteria and were included in the final sample population for analysis (a full description of reasons for exclusion can be found in Figure 1).

### 2.2. Survey Design and Distribution

All surveys were administered using an online survey tool (Qualtrics, Seattle, WA, USA) accessible via hyperlink or QR code. Rather than directly contacting female athletes through email, social media, or word of mouth, these surveys were distributed remotely to NCAA universities located across the United States in which D1 women’s track and field programs were listed on their official athletics websites. All nine regions of the United States were included in the survey distribution process and were contacted via email to their respective compliance officer or equivalent of the women’s track and field programs. By distributing our surveys through athletic staff, we hoped to emulate the process for prescreening athletes at the collegiate level, which would typically require the approval of the coaches or athletic administration. As illustrated in Figure 1, 16 out of 319 universities responded to this survey request, 7 of which agreed to distribute the survey.

Each online survey was divided into 6 subsections. An introduction and purpose statement were included within the first section alongside a standard online informed consent document. If participants consented, they were automatically redirected to the participant demographics section. The demographics section requested information such as age, biological sex, height, weight, race, ethnicity, and pregnancy status. If a participant responded as male, younger than 18 years of age, or indicated that they were pregnant, the survey concluded. Body mass index (BMI) was also calculated using the information gathered in Section 2. The third section of the survey focused on the participant’s competition and sport-related background. In this section participants were asked to list their events (i.e., 400 m hurdles, steeple chase, etc.), the seasons in which they compete, their eligibility status (i.e., freshman, sophomore, etc.), and injury history (i.e., whether an injury was sustained from collegiate participation, and type/classification of injury). This information was important for stratifying each participant into a competition subpopulation, and for understanding injury history. Importantly, participants were also asked a series of questions related to training volume. These questions included weekly mileage, practice training hours, resistance training hours, and cross-training (i.e., combined running and resistance training) hours for both competition (i.e., in-season) and conditioning (i.e., out-of-season) time periods. These training volume variables were later used to estimate total training volumes for each participant, which were then used to assign individuals to a high (≥50th percentile) vs. low (<50th percentile) training volume group (described in more detail in the Statistical Approach section).

The fourth section of the survey contained a triad risk evaluation questionnaire adapted directly from the 12 Triad screening questions outlined by the Female Athlete Triad Coalition in 2016 [2]. These questions related to topics such as body image, dietary habits, menstrual history, and bone health history. An additional 2 questions addressing subjective energy levels during daily life activities as well as during exercise were also incorporated into this section in order to gain an understanding of the athlete’s average subjective energy levels. Notably, energy availability is reportedly difficult to determine outside of a laboratory due to the reliance on estimates rather than direct measures [7]. Therefore, subjective evidence of EA was used as a proxy for objectively measured EA in this study due to the difficulty in obtaining direct measurements. Answers to these questions were later used to determine triad risk for each participant. Because the assessment method of this study was remote, direct reports of certain variables (i.e., Z-scores for determining low BMD) were unable to be gathered. Therefore, triad risk was calculated using individual factors of triad risk, which will be referred to as triad RF counts for this study. These individual factors included self-reports of LEA, low BMI (<18.5 kg/m^2^), incidences of stress reactions/fractions, delayed menarche, and oligomenorrhea and/or amenorrhea. Similar to training volume classifications, participants above the 50th percentile for triad risk factor counts were classified as high risk and participants below the 50th percentile were classified as low risk.

The fifth section of the survey contained the ASSQ Sleep Difficulty Score (SDS) questionnaire, which was included to assess each participant’s sleep quality and normal sleeping habits [8]. Each survey was scored based on a predetermined scoring algorithm, and all scores were then combined to produce an overall SDS. Participants were categorized into SDS groups based on their survey scores: none (0 to 4 points), mild (5 to 7 points), moderate (8 to 10 points), and severe (11 to 17 points) [8].

The sixth section of the survey contained the Perceived Stress Scale (PSS [9]; Cronbach’s alpha reported > 0.7 [10,11]) and the Patient Health Questionnaire (PHQ-9 [12]; Cronbach’s alpha reported > 0.8 [13,14]). The first half of Section 6 was dedicated to the PSS’s 10 questions assessing self-perception of stress levels within the last month. Participants were categorized into PSS groups based on their survey scores: low perceived stress (0 to 13 points), moderate perceived stress (14 to 26 points), and high perceived stress (27 to 40 points) [15]. The second half of Section 6 was dedicated to the PHQ-9. The first 9 questions involved responses to statements describing the degree to which participants had been bothered by certain issues within the last 2 weeks. These issues were related to self-perceptions of pleasure and interest in daily life, perception of energy and concentration levels, and thoughts of self-harm. The final question assessed the difficulty experienced by the participants due to these specific problem statements with regards to whether these difficulties were affecting the overall level of functioning for the participant in activities of daily living. Participants were categorized into PHQ-9 groups based on their survey scores: no symptoms (0 to 4 points), minimal symptoms (5 to 9 points), minor depression/dysthymia/mild major depression (10 to 14 points), moderately severe depression (15 to 19 points), and severe major depression (20 points or greater) [12].

### 2.3. Data Computation and Statistical Approach

Descriptive statistics were performed initially to determine the participant sample. These frequencies and means included age, height, weight, and race. As briefly described in the previous section, training volume was determined through a series of questions in which participants were instructed to provide estimates of average training volume. These questions were then listed as a series of variables (i.e., training mileage, training hours, resistance training hours, and cross-training hours) to be analyzed separately and as a summed total (i.e., total training volume hours) for competition and conditioning seasons, respectively. As alluded to previously, ‘competition’ was used in this study to represent the time periods in which a participant was actively participating in sporting activities (i.e., NCAA-sanctioned track and field meets), commonly referred to as the in-season period. ‘Conditioning’, in contrast, represented the time periods in which a participant was not actively participating in sporting activities and was instead training in preparation for upcoming sports activities, commonly referred to as the ‘off-season’. Additionally, 14 participants replied to training volume questions with a range (e.g., 40–50 miles for competition training mileage). In such a case, the lowest value was recorded in order to standardize these responses for data analysis.

Next, general linear models were used to evaluate the level of association between measures of training volume and total risk factor counts, and binomial logistic regression was used to examine the association between training volume measures and the binomial risk factor classification (i.e., high vs. low risk factor groups). Ordinal logistic regression analyses were also used to evaluate the associations between reported total training volumes and the SDS, PSS, and PHQ-9 categories assigned based on each participant’s response. Independent samples *t*-tests were then used to test for differences in reported training volumes, and Mann–Whitney U tests were used to examine differences in SDS, PSS, and PHQ-9 scores, and in the ordinal responses to each of the 12 questions of the triad risk assessment questionnaire, between the high vs. low training volume groups. Shapiro–Wilk tests were performed to test for the normality of data. All statistical analyses were conducted using the Jamovi software program (version 2.3.21.0), statistical significance was accepted at *p* ≤ 0.050, and data are presented as mean ± standard deviation (s.d.) unless otherwise specified. Of note, this study collected a convenience sample of participants using an online survey design, and therefore, an a priori power analysis was not conducted for this study.

## 3. Results

### 3.1. Demographics and Descriptives

Participant demographics are presented in Table 1 and Figure 2. The total study sample was predominantly comprised of White (85%) distance runners (73%) competing in an NCAA D1 athletic program in the Southeast region of the US (68%). The average age of the sample was 20 ± 2 years with an average BMI of 21.1 ± 2.5 kg/m^2^. As illustrated in Table 1, neither age, weight, height, BMI, or any measure of self-reported training volume was significantly different between multievent athletes and athletes strictly competing in distance events. Also, as shown in Figure 2, the study sample primarily reported low to moderate perceived stress scores (PSS) and minimal to mild depressive symptoms (PHQ-9). Reported sleep difficulty was more variable, with 37%, 24%, and 27% of participants reporting either mild, moderate, or no sleep difficulties (SDS), respectively.

### 3.2. Relationship between Training Volume and Triad Risk

The associations between reported training volume measures and triad risk factor counts are presented in Table 2. As noted previously, general linear models examined the association between reported training volumes (specifically, total weekly mileage, total weekly training hours, total weekly cross-training hours, and total weekly resistance training hours for both the in-season and off-season periods) and the total number of self-reported risk factors based on the 2016 Female Athlete Triad Coalition prescreening survey [2]. These analyses indicated that the only training volume measures that significantly influenced total triad risk factor count were total weekly resistance training hours for the competition (R^2^ = 0.216, β = 0.390, *p* = 0.022) and conditioning periods (R^2^ = 0.180, β = 0.355, *p* = 0.044). No other training volume measure was significantly associated with risk factor count (all *p* ≥ 0.117). Similarly, binomial logistic regression indicated that the only training volume measures that were significantly able to predict whether an athlete would be categorized as high (≥50th percentile for risk factor count) or low (< 50th percentile) triad risk were total weekly competition (odds ratio [OR] = 2.86 [95% CI: 1.06/7.57], *p* = 0.037) and conditioning (OR = 2.38 [95% CI: 1.03/5.50], *p* = 0.042) resistance training hours (Table 2). No other training volume measure was significantly predictive of the triad risk category (all *p* ≥ 0.156).

The results from ordinal logistic regressions evaluating the associations between training volume measures and SDS, PSS, and PHQ-9 categories are presented in Table 3. Similar to triad risk, total weekly resistance training hours for the conditioning period (OR = 0.86 [95% CI: 0.67/1.09], *p* = 0.036) were significantly associated with the SDS category, as was total weekly mileage for the conditioning period (OR = 1.09 [95% CI: 1.03/0.108], *p* = 0.009). Total weekly training hours for the competition (OR = 1.15 [95% CI: 1.02/0.35], *p* = 0.035) and conditioning (OR = 1.18 [95% CI: 1.03/1.38], *p* = 0.031) periods were also associated with PSS category, and total weekly resistance training hours for the competition (OR = 0.408 [95% CI: 0.408 [95% CI: 0.19/0.83], *p* = 0.017) and conditioning (OR = 0.452 [95% CI: 0.22/0.87], *p* = 0.022) periods were predictive of PHQ-9 category. No other significant associations were observed between any other training volume measure and PHQ-9, SDS, or PSS scores (all *p* ≥ 0.172).

These analyses were supplemented further by comparing training volume measures between low and high triad risk groups, the results of which are presented in Figure 3. As seen here, total weekly resistance training hours for the competition (mean diff = 0.722 [95% CI: 0.06/1.39], *p* = 0.34), and conditioning (mean diff = 0.824 [95% CI: 0.09/1.56], *p* = 0.030) periods were significantly higher in the high-risk group compared to the low-risk group. Furthermore, as shown in Figure 4, PSS scores were significantly higher in the high-risk group compared to the low-risk group (mean diff = 5.00 [95% CI: < 0.01/9.00], *p* = 0.038). No other differences were observed between groups (all *p* ≥ 0.061).

## 4. Discussion

Often undetected, the triad is a perilous syndrome known for impacting women of all athletic levels and is therefore screened by most programs before allowing active sports participation [2]. However, there is still much that needs to be uncovered regarding the triad and its development in trained female athletes. Knowing the triad has several screening protocols already in place, such as the 12 questions from the 2016 Coalition triad screening survey [2], this study sought to determine whether self-reported measures of training volume would be associated with self-reported indices of triad risk in female collegiate track and field athletes.

Several measures of self-reported training volume were significantly associated with or predictive of self-reported triad risk. Total weekly resistance training for the competition and conditioning periods were found to significantly influence total triad risk factor count (Table 2), were predictive of triad risk group, and were significantly associated with self-reported depressive symptoms (PHQ-9). Likewise, total weekly mileage and resistance training hours during the conditioning period were associated with self-reported sleep disturbances (SDS), and total weekly training hours for the competition and conditioning periods were significantly associated with self-reported perceived stress (PSS) (Table 3). Lastly, total weekly resistance training hours for competition and conditioning periods and PSS scores were all significantly higher in the high-risk group (Figure 3 and Figure 4). Taken together, these findings suggest that some, but not all indices of self-reported training volume are associated with female athlete triad risk in female NCAA D1 runners.

### 4.1. Resistance Training as a Possible Predictor of Triad Risk

As listed above, resistance training was the main self-reported training measure shown to have significant associations with triad risk. In general, resistance training is beneficial for enhancements in muscular strength, muscular endurance, and flexibility and is recommended to be performed a minimum of two to three times per week for the average adult [16]. For many programs, a certain level of sport-specific resistance training is expected to be completed by athletes, though this baseline can vary by sport, performance level, and sex. For example, a strength-based thrower may be expected to perform a higher volume of resistance training compared to an aerobic endurance-based distance runner. However, like any other exercise modality, resistance training is a metabolically taxing activity, and if not compensated by an equal increase in caloric intake, substantial increases in resistance training volume may contribute to a state of LEA [4,17]. Moreover, increases in resistance training must be supplemented by an adequate protein intake, and prior evidence indicates that females tend to prioritize carbohydrate intake over protein intake, even during periods of increased training [18]. This may be unintentional (e.g., unaware of increased caloric and protein needs) or associated with a compulsion to increase training volume. It is also worth noting that in a sample of female endurance athletes, 30% reported modifying their diets to achieve performance goals [19]. This may be motivated by a desire to decrease body size and increase power-to-weight ratio, both of which are associated with improved performance [20,21].

While not officially identified as a diagnosable mental disease, compulsive exercise addiction is commonly associated with certain eating disorders [22]. When the internalization of an athlete’s body becomes negative or distorted, body dissatisfaction occurs and can result in an innate desire to change one’s appearance, which can then impact physical function and lead to habits such as disordered eating or compulsive exercise [17,23]. A high athletic identity has also been linked with an increased risk for disordered eating and compulsive exercise [24] and could further explain the link between resistance training and triad risk. Moreover, when an athlete has a high athletic identity and has associated their performance with their own self-worth, they are at high risk of continuing to practice while in a state of injury or when chronically under-fueled [24,25]. Ultimately, this internal pressure can lead to compulsive exercise/exercise addiction, leading to an increased risk of injury and LEA [17].

It is also important to recognize that the motivation to change appearance (i.e., appear “slim”) may be external as well. In 2014, Kong and Harris [26] found that over 60% of the elite athletes in their study experienced external pressure from a member of the coaching staff to obtain or maintain a leaner figure at some point in their athletic careers. Therefore, an athlete with a high athletic identity and/or negative body image or body dissatisfaction is likely to spend additional hours on resistance training while also failing to meet optimal nutritional demands, thus leading to LEA and the triad. It should be noted as well that resistance training is still only one part of a female collegiate track and field athlete’s overall energy expenditure during a traditional training plan, and this relationship with triad risk may be predictive of a larger systemic issue with the athlete. This is supported by the observed associations between total training hours and perceived stress (Table 3), and previously reported associations between excessive training hours and increases in depressive symptoms [27].

### 4.2. Mileage as a Predictor of Triad Risk

Unlike resistance training volume, this study found that self-reported training mileage was not a robust predictor of triad risk in female collegiate track and field athletes. Instead, self-reported training mileage only showed a significant relationship with the SDS category (Table 3). As outlined in the limitations section, this was likely due to the inherent limitations of collecting self-reported data. For example, a distance runner could increase or decrease their mileage by 10 miles from 1 week to the next, while training hours (e.g., practice time at the track) and resistance training hours (e.g., practice time in the weight room) will usually be at the same scheduled time for the same amount of time for that entire season. Therefore, mileage may be more variable over the course of a training week, while training hours and resistance training hours are generally planned and more structured. Furthermore, mileage, like the other training volume variables, is also influenced greatly by subject recall. This will be explored in greater detail in the following sections.

Interestingly, other studies that have included measures of training volume have found some evidence that mileage does have a relationship with triad prevalence in female runners. In 2003, Cobb et al. [28] conducted a cross-sectional study in which the associations between disordered eating, menstrual irregularity, and low BMD were evaluated in 91 competitive female distance runners of a similar age range (i.e., 18–26 years old) as our study. In this study, the investigators used a self-administered questionnaire for the assessment of training and menstrual history, a food frequency questionnaire, and the Eating Disorder Inventory (EDI) for the assessment of diet and eating behaviors. Athletes additionally underwent physical clinical assessments, and BMD was recorded via dual-energy x-ray absorptiometry (DEXA). Of the female runners included in this study, the authors reported that 26% were oligomenorrheic and 10% were amenorrheic within the past year. The authors also reported an average of 45% fewer menstrual cycles in the oligomenorrheic and amenorrheic athletes, despite similar demographics and body composition compared with the eumenorrheic athletes. Of note, the athletes with menstrual irregularities in the Cobb et al. study were reported to have performed 18% more mileage on average than that of their eumenorrheic matches over the past 12 months prior to data collection. This would have included both competition and conditioning periods for the population of athletes. Therefore, there is evidence that greater energy expenditure from increased mileage could contribute to greater energy imbalances, which could then lead to the development of the triad via menstrual dysfunctions, especially when considering that athletes with one component of the triad are at a greater risk of developing the remaining components of the triad [2].

### 4.3. Implications for Survey-Based Training Volume as Predictor of Triad Risk

Although more accessible on a broader scale, survey-based triad risk has some inherent limitations that may limit the identification of triad risk in collegiate female athletes. A notable issue in a remote survey-based study is honesty in self-reporting, especially if the distribution of the surveys is in an online, remote fashion where no physical accountability is held for the participants [29]. Even in anonymous surveys, biased perceptions of oneself such as with personal eating and training habits or a fear of divulging information detrimental to social or sporting appearance have the potential to undermine honest responses to these sensitive questions. Faulty recall can additionally weaken accuracy, especially with questions regarding training volume measures and triad risk factors, and certain questions addressing bone health are limited by the fact that many female athletes may not been screened for low BMD or related conditions during general pre-screening due to the limited availability of technology (i.e., DEXA). The estimation of LEA without objective data is also subject to limitations and may have been underestimated by the female athletes in our study population. As found by Capling et al. [30] in a systematic review assessing the validity of certain dietary assessments currently used in athletics, athletes tend to underreport energy intake measures on average by 19%. Therefore, more accessible objective measures may be necessary to reach concrete conclusions regarding triad risk and training volume measures, such as in the 2003 study by Cobb et al. [28]. This could then enable the use of tools such as the Triad Cumulative Risk Assessment [7] to more accurately predict triad risk at prescreening evaluations.

### 4.4. Limitations

This study is not without limitations. As briefly discussed above, self-reporting and subject recall are inherent limitations of online survey-based data collection. Certain questions discussing sensitive topics like disordered eating, menstrual history, and coping habits were likely influenced by participant bias [29] despite reassurances of anonymity. Furthermore, participants were asked to provide average estimates of training volume for competition and conditioning periods rather than direct measurements. The duration of this recall effect may have been substantially increased for certain participants depending on when the survey was taken. Similarly, since this study did not objectively evaluate BMD and LEA using physical collection methods (i.e., DEXA or monitored diets), certain triad risk assessment methods (such as the Triad Cumulative Risk Assessment introduced by the Triad Coalition in 2014 [7]) were unable to be used, and triad risk was estimated instead. However, being that the purpose of this study was to evaluate these associations using a self-report-based method, this was necessary for this study question. Similarly, many NCAA D1 universities do not have the equipment necessary to collect direct measurements of EA and BMD. Lastly, the sample size also presents a significant limitation to the study and also highlights a challenge for the implementation of female athlete triad monitoring. As outlined in Figure 1, of the 319 institutions that were contacted for participation in this survey, only 7 (2.1%) chose to distribute the survey, with the most commonly cited reasons for not distributing surveys being the volume of requests, additional university-specific IRB approval, and athletics board approval. This approach of contacting each institution’s athletic administration directly was based on the notion that the implementation of any female athlete triad monitoring programs would also need to be approved by the administrators and coaches before reaching the athletes. Therefore, this very low response rate may indicate that further outreach efforts are needed to encourage institutional engagement in this very important line of research. It is also important to recognize that the final sample of included responses was from a predominantly White cohort of NCAA D1 distance runners from the southeast region of the United States, which may limit generalizability to the general population of competitive female runners.

### 4.5. Future Directions

To overcome the limitations related to self-reporting described above, future studies may consider supplementing these measures with objective measures of caloric expenditure, caloric intake, and other triad-related indices. By physically screening and collecting direct measures, this method would allow for more accurate body composition and anthropometric measures. Likewise, DEXA or similar methods for measuring BMD could also be incorporated, thus allowing for an increase in precision for prediction of triad risk through the use of multiple triad risk assessment methods such as the 12 Female Athlete Triad Coalition screening questions [2] and the Triad Cumulative Risk Assessment method [7]. Incorporating the use of wearable technologies such as smart watches could be beneficial for overcoming any recall or participant bias limitations with reporting training volume measures as well. This would then result in true averages for training volume measures rather than the recall-dependent estimates used in this study.

Another reasonable direction for future research would be to expand the target population for studies on the prediction of triad risk. Rather than assessing only NCAA D1 female track and field athletes, multiple levels of female athletes could be recruited. These levels could range from recreational to professional and include an increased age range of high school to middle-aged athletes, allowing for an expanded focus on the effects of the triad on development and aging.

## 5. Conclusions

The triad is a significant risk to the health and wellness of female athletes and is generally addressed at pre-screening appointments prior to sports participation. However, many of these screening tools do not include estimates of training volume, which is likely to have an effect on total daily energy expenditure and resultant energy availability for the athletic endeavors of these female athletes. As hypothesized, this study found that there appear to be some training volume measures that were associated with or predictive of triad risk, providing support for further research into the potential predictive value of training volume as an additional marker of triad risk in female runners.

## Figures and Tables

**Figure 1 jfmk-09-00179-f001:**
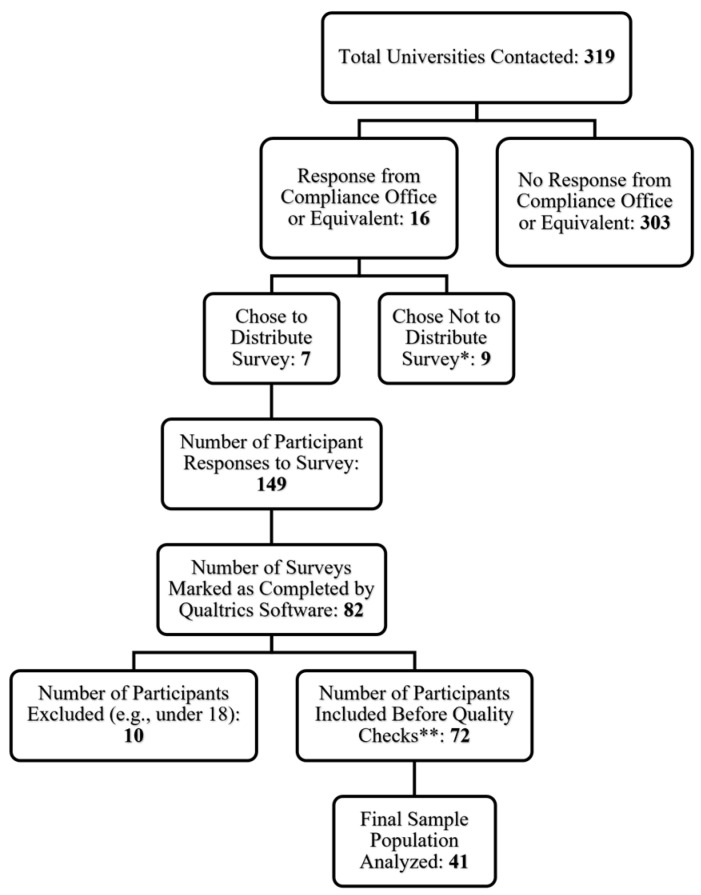
CONSORT Diagram of Survey Completion by Participants. Online surveys (Qualtrics XM, Provo, UT) were remotely distributed by email to compliance offices/equivalence in athletic departments of 319 Division 1 (D1) universities marked as having a National Collegiate Athletic Association (NCAA)women’s track and field program. * The most commonly cited reasons for not distributing surveys to a program’s female athletes were volume of requests, additional university-specific IRB approval, and athletics board approval. ** Quality checks were performed to ensure the participants included in the final sample population had not skipped or left any questions of interest blank.

**Figure 2 jfmk-09-00179-f002:**
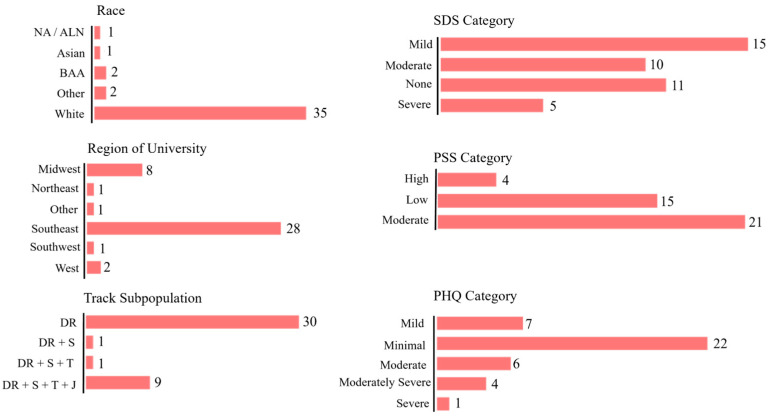
Participant Demographics. Participant distributions are provided across reported racial groups, university regions, track subpopulations, as well as responses to the SDS, PSS, and PHQ-9 assessments. As seen here, the participant sample was predominantly White, was recruited primarily from the Southeast region, mostly comprised of distance runners, and reported mostly moderate to low perceived stress and minimal to moderate depressive symptoms. Responses to the SDS survey were more evenly distributed, with 88% of respondents categorized as reporting none to moderate symptoms of sleeping difficulty. NA/ALN, Native American/Alaskan Native; BAA, Black or African American; DR, distance runners; S, sprinters; T, throwers; J, jumpers; SDS, Athlete Sleep Screening Questionnaire sleep difficulty score [8]; PSS, Perceived Stress Scale [9]; PHQ, Patient Health Questionnaire-9 [12].

**Figure 3 jfmk-09-00179-f003:**
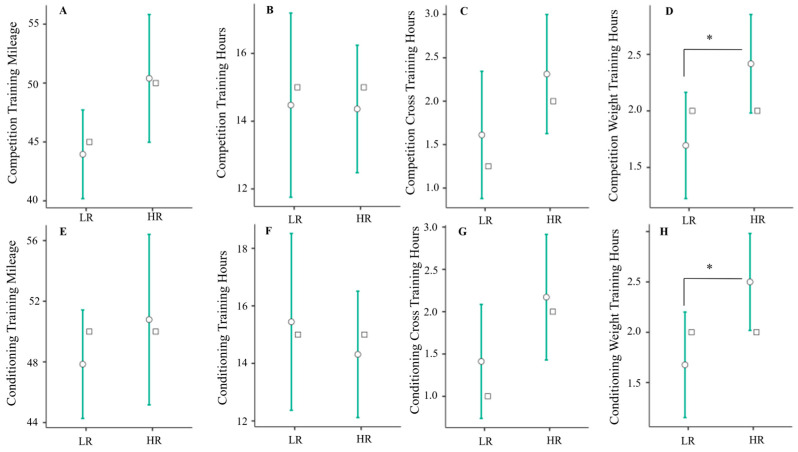
Competition training mileage (**A**); competition training hours (**B**); competition cross training hours (**C**); competition weight training hours (**D**); conditioning training mileage (**E**); conditioning training hours (**F**); conditioning cross training hours (**G**); and conditioning weight training hours (**H**) compared between high and low triad risk groups using independent samples *t*-tests. Competition resistance training hours (**D**) and conditioning resistance training hours (**H**) were both significantly elevated in the high-risk (HR) group compared to the low-risk group (LR). * indicates a statistically significant difference between group means (*p* < 0.050). Error bars represent 95% confidence intervals around the mean, and white squares represent the group median value.

**Figure 4 jfmk-09-00179-f004:**
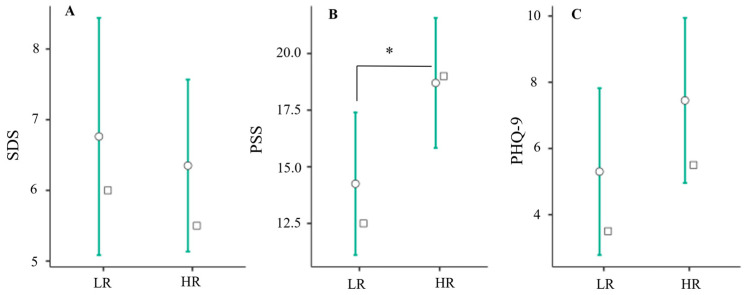
Self-reported sleep difficulty (**A**); perceived stress (**B**); and depressive symptoms (**C**) compared across triad risk groups using a Mann–Whitney U test. Perceived stress (**B**) was significantly elevated in the high-risk (HR) compared to low-risk group (LR). SDS, Athlete Sleep Screening Questionnaire sleep difficulty score [8]; PSS, Perceived Stress Scale [9]; PHQ-9, Patient Health Questionnaire-9 [12]. * indicates a statistically significant difference between group means (*p* < 0.050). Error bars represent 95% confidence intervals around the mean, and white squares represent the group median value.

**Table 1 jfmk-09-00179-t001:** Participant demographics and self-reported training volume measures.

	*n*	Whole Sample (*n* = 41)	Distance Runners (*n* = 30)	Multievent Athletes (*n* = 11)	*p*
Baseline Demographics					
Age (yrs)	40	20 ± 2	20 ± 2	20 ± 1	0.363
Height (cm)	41	164.1 ± 9.2	163.5 ± 9.4	165.5 ± 8.8	0.558
Weight (lb)	40	125.1 ± 14.4	122.7 ± 12.1	132.2 ± 18.6	0.068
BMI (kg/m^2^)	41	21.1 ± 2.5	20.9 ± 2.4	21.6 ± 2.8	0.452
Competition Training Volume Measures					
Total Weekly Mileage	41	47.1 ± 11.1	47.5 ± 10.9	45.9 ± 12.0	0.682
Total Weekly Hours	37	14.4 ± 5.1	14.2 ± 4.8	15.0 ± 6.1	0.680
Total Weekly Cross Training Hours	38	2.0 ± 1.6	1.8 ± 1.6	2.4 ± 1.5	0.351
Weekly Resistance Training Hours	36	2.1 ± 1.0	2.0 ± 1.1	2.1 ± 0.9	0.855
Conditioning Training Volume Measures					
Total Weekly Mileage	39	49.3 ± 10.5	51.1 ± 10.3	44.6 ± 9.9	0.076
Total Weekly Hours	34	14.9 ± 5.7	15.1 ± 5.7	14.4 ± 6.0	0.740
Total Weekly Cross Training Hours	36	1.8 ± 1.6	1.7 ± 1.5	2.1 ± 1.7	0.488
Weekly Resistance Training Hours	35	2.1 ± 1.1	2.1 ± 1.2	2.2 ± 1.1	0.747

*n*, total number of respondents to each question; statistical significance (accepted at *p* < 0.050).

**Table 2 jfmk-09-00179-t002:** General linear models and binomial logistic regression outputs predicting triad risk.

	Linear Regression			Binomial Logistic Regression		
	R^2^	β	*p*	OR	95% CI	*p*
Dependent Variable	Triad Risk Factor Count			Triad Risk Factor Group		
Competition Training Measures						
Total Weekly Mileage	0.077	0.099	0.568	1.05	0.98/1.13	0.156
Total Weekly Hours	0.068	−0.040	0.818	0.988	0.87/1.13	0.860
Total Weekly Cross Training Hours	0.135	0.271	0.117	1.373	0.86/2.21	0.190
Weekly Resistance Training Hours	0.216	0.390 *	0.022	2.86 *	1.06/7.57	0.037
Conditioning Training Measures						
Total Weekly Mileage	0.069	0.058	0.738	1.01	0.95/1.09	0.703
Total Weekly Hours	0.0641	−0.017	0.923	0.963	0.85/1.09	0.552
Total Weekly Cross Training Hours	0.115	0.237	0.176	1.37	0.84/2.23	0.204
Weekly Resistance Training Hours	0.180	0.355 *	0.044	2.38 *	1.03/5.50	0.042

*, statistical significance (*p* < 0.050); OR, odds ratio; Age and race included as a covariate in all regression models.

**Table 3 jfmk-09-00179-t003:** Ordinal logistic regression outputs predicting sleep difficulty, stress, and depressive symptoms.

	OR	95% CI	*p*	OR	95% CI	*p*	OR	95% CI	*p*
Dependent Variable	SDS			PSS			PHQ-9		
Competition Training Measures									
Total Weekly Mileage	1.39	0.76/2.61	0.263	0.98	0.91/1.05	0.597	0.98	0.91/1.05	0.550
Total Weekly Hours	1.07	0.95/1.21	0.272	1.18 *	1.03/1.38	0.031	0.98	0.87/1.11	0.788
Total Weekly Cross Training Hours	1.25	0.82/1.98	0.312	1.26	0.77/2.16	0.374	1.20	0.74/1.92	0.455
Weekly Resistance Training Hours	1.35	0.72/2.58	0.939	1.277	0.65/2.68	0.489	0.408 *	0.19/0.83	0.017
Conditioning Training Measures									
Total Weekly Mileage	1.09 *	1.03/1.18	0.009	1.00	0.93/1.07	0.975	1.04	0.96/1.12	0.338
Total Weekly Hours	1.08	0.97/1.22	0.172	1.15 *	1.02/1.35	0.035	0.99	0.88/1.11	0.858
Total Weekly Cross Training Hours	0.927	0.62/1.39	0.710	1.24	0.74/2.17	0.423	0.93	0.56/1.49	0.753
Weekly Resistance Training Hours	1.85 *	1.05/3.39	0.036	1.28	0.68/2.51	0.454	0.452 *	0.22/0.87	0.022
Total Risk Factor Count	0.86	0.67/1.09	0.217	1.11	0.86/1.45	0.442	1.17	0.92/1.49	0.211

*, statistical significance (*p* < 0.05); OR, odds ratio; Age and race included as covariates in all regression models.

## Data Availability

The data from this study will be make available upon reasonable request and in accordance with the policies of The University of Southern Mississippi.

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
