# Peer review of "Preliminary Evaluation of Self-Reported Training Volume as an Adjunct Measure of Female Athlete Triad Risk in Division 1 Collegiate Female Runners"

_jfmk, 2024, doi:10.3390/jfmk9040179_

Round 1

Reviewer 1 Report

Comments and Suggestions for Authors

Thanks to the Editor for the opportunity to review this manuscript.

The work is well written, the rationale is well contextualized and supported by adequate literature, and the limitations are well identified; this makes the presentation of the results, which are certainly not incontestable, intellectually honest. However, i would like the following points to be addressed.

Abstract: 

- Even though it is a well-known acronym, it would be good to add the extended form of NCAA.

-It would be good to maintain consistency in the way you write numbers (letters or digits).

2.2 Survey Design and Distribution:

- The section is well written and comprehensive. However, to improve readability, I would suggest to lighten it by shortening and avoiding any unnecessary or redundant information.

2.3 Data Computation and Statistical Approach:

- I see you used both parametric and non-parametric tests. How did you choose these tests? What normality tests did you perform? 

4.4. Limitations and Future Directions

- I would change the name of the section to "Limitations".

5. Conclusions

- I would avoid to repeat the purpose of the work.

Reviewer 2 Report

Comments and Suggestions for Authors

Abstract: when stating the derived p-value, simply state p=.044 or p=.037, etc. 

line 30: Replace "Over recent years,..." with "In 2014, the triad was further categorized....."

Line 55: remove "Interestingly"

Otherwise, I find no missing discussion points, flawed methodology, or writing style concerns.

* What is the main question addressed by the research?
The main question of the researchers' study was, "Is self-reported training volume predictive of female athlete triad risk in NCAA Division 1 female track and field athletes.

* Do you consider the topic original or relevant to the field? Does it address a specific gap in the field? Please also explain why this is/ is not the case.
The above question is timely and relevant given the continued need to address women's health issues in this group of young women athletes. The use of a questionnaire in this manner (anonymous, self-report) has potential use by individual programs in the future where specific athlete risk factors may be identified and mitigated.

* What does it add to the subject area compared with other published material?
The multi-faceted (adding training volume) questionnaire adds depth and specificity to previously published material; the surveys as part of this study still include relevant variables such as dietary habits, menstrual health, etc. while including training volume which is many times difficult to obtain - or is obtained separately on different participants.

* What specific improvements should the authors consider regarding the methodology? What further controls should be considered?
The methodology is sound; the researchers address the limitation regarding survey response rate and the homogeneity of the respondents. The results are still a publishable step in the right direction for this research area.

* Are the conclusions consistent with the evidence and arguments presented and do they address the main question posed? Please also explain why this is/is not the case.
The results are consistent with the evidence and arguments provided in the Introduction and they address the main question identified above. The authors have written a non-bias, well-balanced discussion of their results and have not over-interpreted their meaning and application.

* Are the references appropriate?
Yes.
* Any additional comments on the tables and figures.
The tables and figures are appropriate; in particular the figures facilitate digestion of the results in an efficient manner and they are non-bias.

Round 2

Reviewer 1 Report

Comments and Suggestions for Authors

Many thanks to the authors for their efforts and excellent work. The paper is clearer, the readability is improved and some important points, such as the statistical analysis, are now more comprehensive. My opinion is positive and, as far as I'm concerned, the work is worthy of being published.